# Design and Validation of a Process Based on Cationic Niosomes for Gene Delivery into Novel Urine-Derived Mesenchymal Stem Cells

**DOI:** 10.3390/pharmaceutics13050696

**Published:** 2021-05-11

**Authors:** Yerai Vado, Gustavo Puras, Melania Rosique, Cesar Martin, Jose Luis Pedraz, Shifa Jebari-Benslaiman, Marian M. de Pancorbo, Jon Zarate, Guiomar Perez de Nanclares

**Affiliations:** 1NanoBioCel Research Group, Laboratory of Pharmacy and Pharmaceutical Technology, Faculty of Pharmacy, University of the Basque Country UPV/EHU, 01006 Vitoria-Gasteiz, Araba, Spain; yerai.vado@ehu.eus (Y.V.); gustavo.puras@ehu.eus (G.P.); joseluis.pedraz@ehu.eus (J.L.P.); jon.zarate@ehu.eus (J.Z.); 2Rare Diseases Research Group, Molecular (Epi) Genetics Laboratory, BioAraba Health Research Institute, Araba University Hospital-Txagorritxu, 01009 Vitoria-Gasteiz, Araba, Spain; 3BIOMICs Research Group, Microfluidics Cluster UPV/EHU, Lascaray Research Center, University of the Basque Country UPV/EHU, 01009 Vitoria-Gasteiz, Araba, Spain; melania.rosique@gmail.com (M.R.); marianpancorbo@gmail.com (M.M.d.P.); 4Biofisika Institute (UPV/EHU, CSIC), Department Biochemistry and Molecular Biology, University of the Basque Country University (UPV/EHU), 48940 Leioa, Bizkaia, Spain; cesar.martin@ehu.eus (C.M.); shifa.jebari@ehu.eus (S.J.-B.)

**Keywords:** urine-derived mesenchymal stem cells, primary cell culture, nonviral vectors, cationic niosomes, gene therapy

## Abstract

Background: Mesenchymal stem cells (MSCs) are stem cells present in adult tissues. They can be cultured, have great growth capacity, and can differentiate into several cell types. The isolation of urine-derived mesenchymal stem cells (hUSCs) was recently described. hUSCs present additional benefits in the fact that they can be easily obtained noninvasively. Regarding gene delivery, nonviral vectors based on cationic niosomes have been used and are more stable and have lower immunogenicity than viral vectors. However, their transfection efficiency is low and in need of improvement. Methods: We isolated hUSCs from urine, and the cell culture was tested and characterized. Different cationic niosomes were elaborated using reverse-phase evaporation, and they were physicochemically characterized. Then, they were screened into hUSCs for transfection efficiency, and their internalization was evaluated. Results: GPxT-CQ at a lipid/DNA ratio of 5:1 (*w*/*w*) had the best transfection efficiency. Intracellular localization studies confirmed that nioplexes entered mainly via caveolae-mediated endocytosis. Conclusions: In conclusion, we established a protocol for hUSC isolation and their transfection with cationic niosomes, which could have relevant clinical applications such as in gene therapy. This methodology could also be used for creating cellular models for studying and validating pathogenic genetic variants, and even for performing functional studies. Our study increases knowledge about the internalization of tested cationic niosomes in these previously unexplored cells.

## 1. Introduction

Gene therapy is defined as the process of transferring external genetic material into a cell, tissue, or whole organ with the aim of curing a disease or improving the clinical status of a patient [1]. It is a field of emerging importance for the research and treatment of various types of diseases [2].

Mesenchymal stem cells (MSCs) are a pool of multipotent adult stem cells present in a variety of niches that can differentiate into mesoderm-derived cells, such as osteocytes or adipocytes [3]. Several studies have shown that MSCs can also differentiate into cells of endodermal or ectodermal origin [4]. In addition to in vitro differentiation into osteocytes, adipocytes, and chondrocytes, MSCs in culture can attach to plastic and grow under standard culture conditions. Moreover, these cells must present CD73, CD90, and CD105 surface antigens and lack expression of CD14, CD34, and CD45 [5].

MSCs are beneficial to work with due to the ease of obtaining them and the few ethical issues they present. They are self-renewable and multipotent, and they can be expanded in culture with great genomic stability. Taking everything into account, mesenchymal stem cells are important in cell therapy, regenerative medicine, and tissue repair [6,7]. In addition, as cells are obtained from an autologous source, they do not promote an immune response or induce rejection [8,9].

MSCs can be isolated from a wide variety of adult tissues, but those that are most studied and used are derived from bone marrow [3], adipose tissue [10], and umbilical cord [11]. The disadvantage of obtaining the mentioned cells is that the sources are limited and the procedures to obtain them are invasive [8,12]. This is why a more convenient novel source of mesenchymal stem cells is needed. Zhang et al. described the first culture of MSCs isolated from human voided urine (hUSC) [13] and, since then, researchers have focused their attention on these cells. As other MSCs, hUSCs show an appropriate proliferation capacity and have multilineage differentiation potential [12]. Compared to other MSCs, hUSCs can be isolated in a noninvasive, simple, reliable, and safe way using a low-cost methodology. In addition, they have great potential to be used for clinical applications, and they can also be used as biological models for pharmacology and toxicology tests [14]. These human-derived cells can be used in a personalized way with high-throughput screening in predictive toxicology. hUSCs have the potential to be used in applications for regenerative (for example, in urological tissue engineering [15], bone regeneration [16], and induction of iPSCs with different purposes such as neuroregeneration [17] or dental reconstruction [18]) and personalized medicine. In addition, they can be used in gene therapy approaches for treating neurological and blood disorders or vascular and musculoskeletal diseases as well as to impact tumor growth [19]. Some authors also described these cells as a novel biological resource for the discovery of new drugs [14]. Using hUSCs as disease models overcomes the drawback that animal models are not always a precise model and present ethical issues [14]. Furthermore, they can be obtained easily and noninvasively [12,13,20,21]. For these reasons, we were interested in developing a protocol to isolate and characterize hUSCs and to transfect them with nioplexes for the purpose of application in gene editing.

In the same way as MSCs, nonviral vectors (especially niosomes) are gaining attention in gene therapy [7].

Viral vectors have been successfully used in numerous clinical trials, but their application has been linked to carcinogenesis [22] and immunogenicity problems [23], in addition to having limited DNA-packaging capacity [24] and being challenging to produce [25]. Nonviral vectors, on the other hand, present important advantages such as the size of the nucleic acid to transfect, which is theoretically unlimited, and they are easier and cheaper to produce [26,27]. However, the duration of gene expression is an issue that nonviral vectors have to face, as well as their low transfection efficiency [28]. Among the nonviral vectors, cationic liposomes are among those that are most employed and studied [29].

Niosomes are emerging nonviral drug delivery systems with a bilayer structure that can be used as an alternative to liposomes. While liposomes are composed of phospholipids, niosomes contain nonionic surfactants [30,31]. Some interesting properties of niosomes for gene delivery applications include, for instance, their low toxicity and cost, good chemical stability, and large DNA-packaging capacity [32]. In any case, such properties depend on both the composition of the niosome formulation and the method for their production.

Niosomes can incorporate cationic lipids for gene delivery purposes; thus, via simple electrostatic interactions, nucleic acids bind to the vesicle and form complexes known as nioplexes [33].

Essentially, niosomes have three main components: (1) nonionic surfactants, which increase the stability of niosomes [34,35], (2) helper lipids that enhance the physicochemical properties of the lipid emulsion, as they can modify the morphology, permeability, storage time, nucleic acid release, and stability of the niosome [35,36], and (3) cationic lipids that interact electrostatically with the nucleic acid to form nioplexes and whose properties influence the transfection efficiency and toxicity [33,35].

In summary, the aim of this work was to isolate human urine-derived mesenchymal stem cells and to develop an easy protocol for transfecting them using nonviral vectors. Applications of this procedure could include its use in gene editing, functional studies of novel variants identified at candidate genes.

## 2. Materials and Methods

### 2.1. Cell Culture Isolation

For isolation of the hUSCs, the protocol of Chen et al. [37] was followed. Once cells were isolated, various conservation methods were tested for posterior cell recovery. The influence of culture medium was also evaluated (see Appendix A).

### 2.2. Cell Characterization

For cell characterization, cell surface markers were analyzed by flow cytometry. Briefly, 50,000 cells were grown in 24-well culture plates. For this assay, not only were cells isolated from urine examined but commercial mesenchymal stem cells (PCS-500-011™, ATCC, Manassas, VA, USA) were also used as a positive control while HEK293 cells (ATCC) were used as a negative control.

When cell confluence was approximately 80%, cells were fixed with 200 μL paraformaldehyde (PFA; PanReac AppliChem, Barcelona, Spain) for 10 min. Once fixed, cells were scraped and centrifuged at 3000 rpm for 5 min. The supernatant was discarded, and the pellet was resuspended in 50 μL blocking solution, which was a solution of 5% inactive FBS (Gibco-Thermo Fisher Scientific, Waltham, MA, USA) in PBS. Cells were kept in this solution for 30 min at room temperature. Then, fluorescent antibodies were added at a final dilution of 1/100, and cells were incubated for 45 min at room temperature in the dark. The antibodies and their fluorophores are listed below, all of which were acquired from Sigma- Aldrich (Saint Louis, MO, USA):•Surface markers specific for mesenchymal stem cells: antiCD73-AlexaFluor488, antiCD90-APC, and antiCD105-PE;•Surface markers that are absent in mesenchymal stem cells: antiCD14-FITC, antiCD34-APC, and antiCD45-PE.

For verifying the binding specificity of the antibodies, isotype controls for each fluorophore were also used. As the isotypes do not recognize any protein, every signal obtained from them would be due to unspecific binding. This way, background staining levels can be determined [38]. Once the incubation was completed, unbound antibody was removed by centrifugation of cells at 3000 rpm for 5 min, and the supernatant was discarded. The pellet was resuspended in 200 μL of PBS and the sample was introduced to a CytoFLEX (Beckman Coulter, Brea, CA, USA) flow cytometer. The results were analyzed using CytExpert software (Beckman Coulter, v. 2.3.0.84, Brea, CA, USA).

### 2.3. Plasmid Propagation and Elaboration of Nioplexes

Four different cationic niosomes, which presented the best results in previous studies carried out in the laboratory [39,40], were prepared using the reverse-phase evaporation. For the niosome named GPxT-CQ 2.5 mg (0.05% *w*/*v*), chloroquine diphosphate (Sigma-Aldrich, St. Louis, MO, USA) was dissolved in milliQ water in order to obtain the aqueous phase. In parallel, the organic phase was prepared as follows: 5 mg (0.1% *w*/*v*) lipid, 12.5 mg (0.25% *w*/*v*) Poloxamer^®^ 407 (Sigma-Aldrich), and 12.5 mg polysorbate 80 (Sigma-Aldrich) were dissolved in 1 mL dichloromethane (PanReac). The DLT60 niosome was prepared as described by Mashal et al. [40]. The aqueous phase of the DST20 niosome was composed of a 5 mL solution of 0.49% polysorbate 20 (Sigma-Aldrich) in water. In the organic phase, 6.7 mg of 1,2-di-*O*-octadecenyl-3-trimethylammonium propane (DOTMA; Avanti, Alabama, USA) and 19 µL of squalene (Sigma-Aldrich) were dissolved in 1 mL of dichloromethane. Lastly, a formulation named N4 was manufactured. For this niosome, in the aqueous phase, 25 mg of polysorbate 80 was dissolved in 5 mL of distilled water. In the organic phase, 5 mg of lipid and 20 μL of squalene were dissolved in 1 mL of dichloromethane. Once the different phases were prepared, the aqueous phase was added to the organic phase, and the emulsion was obtained via the sonication of the mixture for 30 s at 50 W (Branson Sonifier 250^®^, Danbury, CT, USA). The organic solvent was then evaporated under magnetic stirring for 45 min at room temperature, leaving the cationic niosomes in the aqueous medium. The resulting niosomes were at a concentration of 1 mg cationic lipid/mL.

Previously described protocols (Mashal et al., 2017) were used for the propagation, purification, and quantification of pCMS-EGFP plasmid (5541 bp, Plasmid Factory, Bielefeld, Germany). The nioplexes were produced by mixing an appropriate volume of pCMS-EGFP plasmid stock solution (0.5 mg/mL) with different amounts of the cationic niosome suspension (1 mg/mL) to obtain different cationic niosome/DNA mass ratios (*w*/*w*). To improve the electrostatic interaction between the cationic niosome and the negatively charged DNA, the mixture was left for 30 min at room temperature.

### 2.4. Physicochemical Characterization of Cationic Niosomes/Nioplexes

First, 50 μL of the samples were dispersed in 950 μL of 0.1 mM NaCl solution, and all measurements were carried out in triplicate using a Zetasizer Nano ZS (Malvern Instruments, Malvern, UK). Dynamic light scattering (DLS) was used to determine the particle size and polydispersity index (PdI). Particle size was determined by cumulative analysis of the recorded hydrodynamic diameter. Laser Doppler Velocimetry (LDV) was used to calculate the zeta potential of particles.

The morphology of the cationic niosomes was assessed by transmission electron microscopy (TEM). Briefly, to perform negative staining, a 5 μL sample was adhered onto glow-discharged carbon-coated grids for 90 s, after which the grid with sample was settled onto a drop of 1% uranyl acetate for another 90 s. The samples were examined under TEM using a JEOL JEM 1400-Plus (JEOL Ltd. Akishima, Tokyo, Japan), in bright-field image mode using an accelerating voltage of 120 kV.

### 2.5. Cell Culture and In Vitro Transfection Assays

hUSCs were seeded in 24-well culture plates at a density of 50,000 cells/well with 300 μL complete medium without antibiotics. After 24 h (70–80% confluence), the medium was removed, and cells were washed with serum-free Opti-MEM^®^ solution (Gibco-Thermo Fisher Scientific). Then, 250 μL nioplex solution (1.25 μg DNA) diluted in serum-free Opti-MEM^®^ was added to the cells at different cationic lipid:DNA mass ratios (*w*/*w*). Cells were left for 4 h at 37 °C. After the incubation time, the transfection medium was removed, and complete medium without antibiotics was added. At this moment, cells were incubated for further 48 h. Following this incubation period, both transfection efficiency and cell viability were determined. Qualitative analysis was performed using an inverted microscope equipped with the EclipseTE2000-S attachment (Nikon, Tokyo, Japan) for fluorescent observation. For quantitative determination, FACSCalibur flow cytometer analysis (Becton Dickinson Biosciences, San Jose, CA, USA) was performed.

To analyze cell viability by flow cytometry, cells were stained with propidium iodide (Sigma-Aldrich). The FL1 (530/30) detector was used to detect EGFP-expressing transfected cells, and the FL3 (670) detector was used to detect dead/dying cells. Experiments with untransfected cells were used as negative controls, and Lipofectamine™ 2000 (Invitrogen, CA, USA) was used as a positive control. As a minimum, 10,000 gated events were collected and analyzed for each sample using the BD CellQuest™ Pro Software (Becton Dickinson Biosciences). Each condition was analyzed in triplicate.

### 2.6. Cell Uptake and Intracellular Distribution of Nioplexes

For the cellular uptake assay, cells were transfected as described above; however, in this assay, FITC-labeled pCMS-EGFP (DareBio, Madrid, Spain) was used. After 4 h of incubation with the vectors at 37 °C, the transfection medium was removed and cells were washed with PBS, detached, and analyzed using a FACSCalibur flow cytometer (Becton Dickinson Biosciences) with the FL1 channel. For each sample, 10,000 events were analyzed. Data are shown as the percentage of FITC-positive cells. Nontransfected cells were used as a negative control. Each condition was analyzed in triplicate.

### 2.7. Intracellular Trafficking Studies

The endocytosis mechanisms involved in the uptake of nioplexes were evaluated by the colocalization of nioplexes (prepared with FITC-pCMSEGFP) with different fluorescently labeled endocytosis markers, all obtained from Invitrogen (Carlsbad, CA, USA). hUSC cells were seeded on coverslips on 24-well culture plates at a density of 80,000 cells/coverslip and transfected with the nioplexes containing the FITC-labeled pCMS-EGFP plasmid for 3 h. After this time, different endocytic vesicle markers were added and incubated for an additional hour with either AlexaFluor^®^ 594-Cholera Toxin (10 μg/mL), AlexaFluor^®^ 568-Transferrin (50 μg/mL), 8.33 μM AlexaFluor^®^ 568-labeled dextran, or Lysotracker (140 nM), which are markers for clathrin-mediated endocytosis (CME), caveolae raft-mediated endocytosis (CvME), the macropinocytosis pathway, and the late endosomal compartment, respectively [41,42]. Cells were fixed with 4% PFA and mounted with Fluoroshield™ with DAPI (Sigma-Aldrich) for examination by confocal laser scanning microscopy (CLSM) using a Zeiss Axio Observer with Apotome 2 (Zeiss, Oberkochen, Germany). In confocal micrographs, the colocalization of the nioplexes (green) and the endocytic pathway (red) was shown as a yellow signal. Fiji ImageJ software was used to analyze the images. The analysis of colocalization was performed using the cross-correlation function (CCF). The colocalization of the green and red signal was analyzed using the Fiji ImageJ software (National Institute of Health, Bethesda, MD, USA, 1.52p version) and quantified by cross-correlation analysis as described in previous reports [43]. Briefly, the green signal image was shifted in the *x*-direction pixel by pixel relative to the red signal image, and the respective Pearson’s coefficient was calculated, which was then plotted as a function of the pixel shift (δx), thereby obtaining the cross-correlation function (CCF). Colocalizing structures peaked at δx = 0 and presented a bell-shaped curve.

## 3. Results and Discussion

### 3.1. Cell Culture Isolation

After 2 weeks of the initial seeding of the urine, groups of cells were present in the wells. They could be passed into a flask and, soon, a homogeneous population was obtained. As fibroblasts, hUSCs were spindle-shaped and had an approximate size of 100 μm (Figure 1).

### 3.2. Cell Characterization

Cell populations analyzed by flow cytometry showed the surface antigen expression pattern described in Table 1. On the one hand, for the three markers that should be present (CD73, CD90, and CD105), cells showed high fluorescence levels. On the other hand, for those that should be absent (CD14, CD34, and CD105), there was no observable signal due to the corresponding antibodies. It must be highlighted that the antiCD45-PE antibody displayed more fluorescence than expected. Nevertheless, it did not reach the values of the positive antigens. As suspected, HEK293 cells were not positive for MSC markers.

Compared to the reference values proposed by the International Society for Cellular Therapy (ISCT) [5], this culture did not fulfill the hallmarks. However, this is not surprising as, in the literature, it has been demonstrated that conclusive characterization of this cell type requires more in-depth analysis. Moreover, results obtained from different groups are not concordant, and the expression levels established by the ISCT are not always obtained [14]. Moreover, commercial cells, which are supposed to meet quality criteria, did not reach the minimum required either, and their percentages were similar to those of hUSCs.

### 3.3. Physicochemical Characterization of Cationic Niosomes/Nioplexes

The results for analysis of the GPxT-CQ cationic niosome and the nioplex at a cationic lipid:DNA ratio of 5:1 are summarized in Table 2. When DNA was incorporated into the cationic niosome, the size of the particles increased from 110 to 162 nm. This was expected as the size must increase upon plasmid complexing in the formulation. The values in terms of nanoparticle size remained on the nanometric scale, making them adequate for gene delivery [44].

Regarding the zeta potential, when DNA was complexed to the cationic niosome, it decreased from 33.4 ± 5.7 to 21.2 ± 2.4 mV (Table 2). The high positive charge of GPxT-CQ (>+25 mV) makes it appropriate for complexing with a negatively charged nucleic acid prior to cellular internalization [45]. Furthermore, the positive charges of the cationic niosome, which has cationic lipids, are partially neutralized by the negative charge of the nucleic acid. In addition, this positive zeta potential helps in the interaction between the formulation and the negatively charged cell membrane [46]. Taking all this into account, it remains clear that an interaction occurred between pCMS-EGPF and GPxT-CQ.

Measurements of both the naked cationic niosome and the nioplexes presented low values of polydispersity index (PdI) (0.13 ± 0.01 and 0.31 ± 0.04, respectively). A higher value of PdI was observed in the nioplex than in the naked cationic niosome. As in the nioplexes the zeta potential was less positive due to the presence of DNA (<+25 mV), whereby the repulsion forces were not big enough, and some of the particles tended to aggregate due to interactions between particles, such as van der Waals or hydrogen bonding [47].Moreover, small polydispersity values usually enhance gene delivery by vehicles [48].

As illustrated in Figure 2, the GPxT-CQ formulation showed a spherical morphology and multilamellar structure when studied by TEM, and no aggregates were present. Usually, lipidic nanoparticles are spherical [49]. However, many formulations with cationic lipids used for gene therapy form lamellar structures when complexed with DNA [50], as was the case with the nanoparticle used in this work (Figure 2).

The use of nonlamellar vesicles is not essential for delivering genetic material into the cell [51]. Moreover, in some studies, it has been demonstrated that lipidic nanoparticles with a multilamellar structure show enhanced gene transfection efficiency. In determining and analyzing the nucleic acid–nanoparticle structure, a rational approach to design could be employed to achieve better gene delivery [52]. Taking this into account, the formulation used in this work presents a promising physical structure for efficient gene delivery into mesenchymal stem cells.

### 3.4. Cell Viability and Transfection Efficiency of Nioplexes

The screening of different formulations at different cationic lipid:DNA ratios showed that the one with the best transfection efficiency without compromising viability was GPxT-CQ at a cationic lipid:DNA ratio of 5:1 (Figure 3). The transfection efficiency was 6.81% ± 0.8% and the cell viability was 77.5% ± 4.7%. Compared to the commercially available Lipofectamine^TM^ 2000 reagent, the GPxT-CQ cationic niosome formulation at a cationic lipid:DNA ratio of 5:1 showed a higher percentage of transfected cells (6.81% vs. 0.33%), with similar cell viability values (77.5% vs. 83.8%); thus, it was used for further experiments.

Cell transfection efficiency was not very high for GPxT-CQ or the positive control Lipofectamine (6.81% ± 0.8% vs. 0.33% ± 0.06%, respectively). This could be explained by the fact that primary mesenchymal stem cells are difficult to transfect and there is not yet an efficient method for delivering genetic material into them [53,54,55]. Even though transfection efficiency can be improved, a high value is not needed for every purpose, as there are studies in which transfection efficiencies were similar to those obtained in this work [56]. Besides, the transfection efficiency for GPxT-CQ was better than that obtained with Lipofectamine; accordingly, the synthetized formulation is advantageous compared to Lipofectamine. Upgraded gene delivery was obtained in some cases by complexing the genetic material to low-molecular-weight chitosans [57]. The polar head groups of the cationic lipids have also been described as a key factor affecting the transfection efficiency [58].

It has been described that chloroquine shows toxicity that could limit cell viability and, consequently, endanger clinical applications [59]. However, in our case, cell viability was not highly compromised as in cells transfected with GPxT-CQ, the percentage was around 77%, suggesting that the low transfection could be associated with cellular uptake of the nioplexes.

### 3.5. Cellular Uptake Studies

The GPxT-CQ cationic niosome conjugated with DNA at ratio 5:1 was used to assess uptake in hUSC cells. As shown in Figure 4, GPxT-CQ had an uptake percentage of 15.16% ± 1.07%, lower than that of Lipofectamine, which was 31.26% ± 1.76%. As can be seen in the microscopy images in Figure 4, cells treated with GPxT-CQ maintained their morphology and looked healthier.

### 3.6. Trafficking of the Nioplex

The transfection efficiency can be directly affected by the internalization pathway. Thus, trafficking studies were performed with the aim of clarifying the transfection process in hUSCs and to understand why the transfection efficiency was low despite cellular uptake. Cells generally use endocytic pathways to internalize nonviral vectors, primarily clathrin-mediated endocytosis (CME), caveolae-mediated endocytosis (CvME), and micropinocytosis [60,61]. Additionally, each pathway affects the effectiveness of DNA release and its performance inside cells [62]. The intracellular internalization of the selected nioplexes in hUSCs is represented in Figure 5. The main colocalization between the endocytosis pathway and the GPxt-CQ-based nioplexes occurred via caveolae-mediated endocytosis (CvME), where the mean CCF peak value was 0.36% ± 0.004%. Macropinocytosis was also an important pathway, as the CCF peak value was 0.24% ± 0.005%. The CCF value was almost zero with Lysotracker. These results are in accordance with the theory postulating that the CvME route avoids lysosomes and, thus, they do not integrate into late endosomes. Macropinocytosis is also believed to follow the nonacidic and nondigestive route [63,64]. Regarding the effect of chloroquine, it was previously reported that chloroquine prevents endosomal acidification as well as inhibits lysosomal enzymes that could damage the genetic material [65,66], which could explain, at least in part, the results obtained in the screening process.

In any case, such results obtained with specific endocytosis markers could be further completed with the use of appropriate and specific inhibitors of main endocytosis pathways, such as genistein to inhibit the caveolae-mediated pathway, chlorpromazine to inhibit the clathrin-mediated pathway, methyl-β-cyclodextrin to inhibit both pathways, or wortmannin to inhibit macropinocytosis-mediated pathways [67,68,69].

## 4. Conclusions

In this work, we highlighted that urine can be used as an easily accessible source of mesenchymal stem cells. In addition, these cells can be used for nonviral gene delivery experiments for future clinical gene therapy purposes in genetic disorders by modifying the mutated genome with technologies such as CRISPR/Cas9. However, the process currently has low transfection efficiency due to the difficulty in transfecting mesenchymal stem cells and the lack of a standardized and efficient protocol.

Despite the transfection efficiency not being high, GPxT-CQ at a cationic lipid:DNA ratio of 5:1 was an adequate nioplex for gene therapy in hUSCs and gave promising results. Furthermore, depending on the use, high rates of transfection may not be necessary. Therefore, the clinical application of these nioplexes combined with hUSCs cannot be discarded. This formulation was integrated into the cell via CvME and it did not interact with lysosomes. In the future, further experiments should be carried out in order to obtain better uptake percentages and more effective internalization of the DNA into the cell nucleus, for example, following such strategies as the use of polymers such as oligochitosans. Despite primary cells being difficult to transfect, the process could also be optimized to achieve better transfection rates. Nevertheless, it must be reiterated that these results do not limit the clinical applicability of this process.

In summary, this study describes the first protocol for obtaining and transfecting urine-derived mesenchymal stem cells with nonviral vectors. Thus, it opens the door to using this gene delivery process for clinical purposes.

## Figures and Tables

**Figure 1 pharmaceutics-13-00696-f001:**
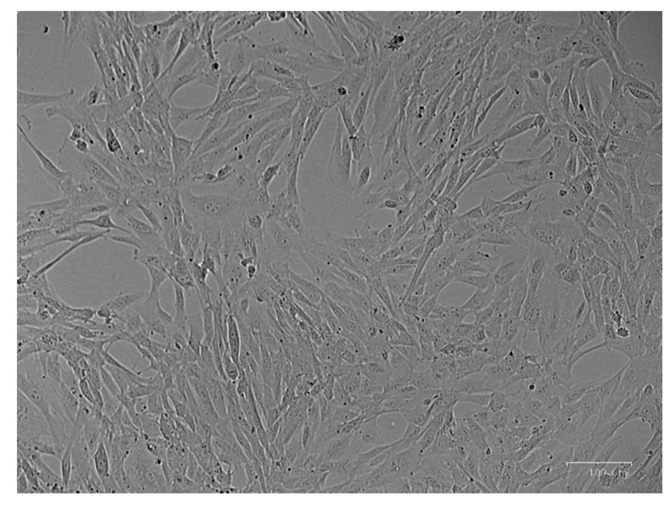
Image of the culture of hUSCs at passage 2 taken using a ZOE™ Fluorescent Cell Imager. Homogeneous spindle-shaped cells are shown. Scale bar: 100 μm.

**Figure 2 pharmaceutics-13-00696-f002:**
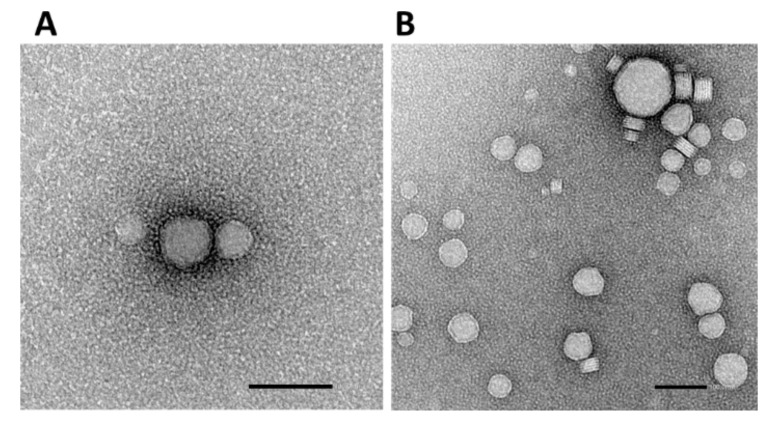
TEM images of GPxT-CQ cationic niosome. Scale bar: 100 nm. (**A**) 80,000× magnification; (**B**) 50,000× magnification.

**Figure 3 pharmaceutics-13-00696-f003:**
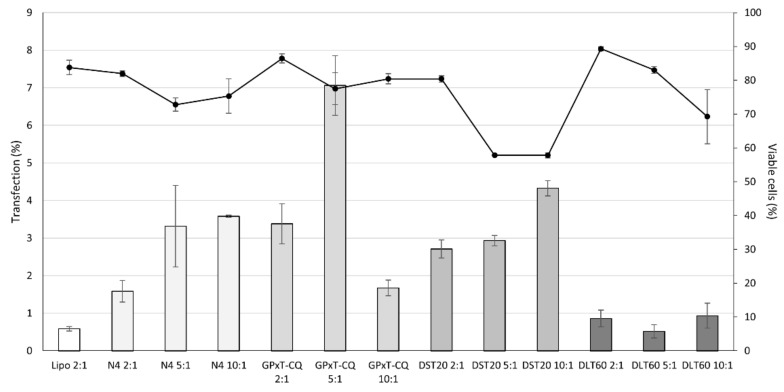
Screening of different nioplexes (N4, GPxT-CQ, DST20, and DLT20) at different cationic lipid/DNA ratios. The bars represent the transfection percentage measured by flow cytometry. Dots joined by the black line show cell viability. Each value represents the mean ± SD (*n* = 3).

**Figure 4 pharmaceutics-13-00696-f004:**
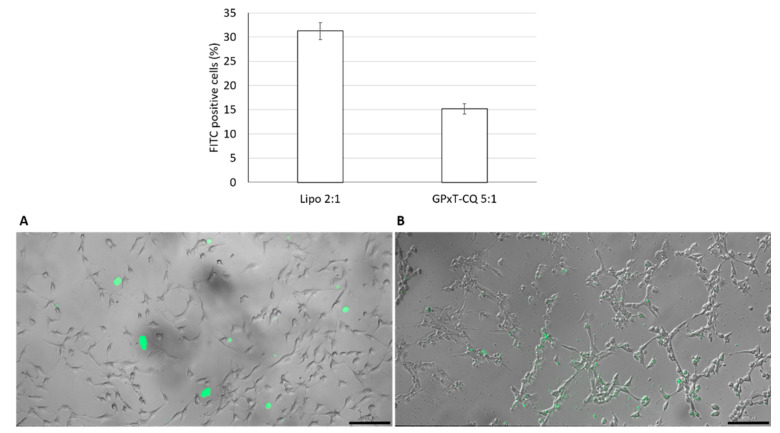
Uptake of FITC-labeled nioplexes in hUSC cells. Lipofectamine was added at a mass ratio of 2:1 and GPxT-CQ 5:1 (*w*/*w*). Upper panel: percentage of FITC-positive cells and viability. Data represent the mean ± SD (*n* = 3). Lower panels: fluorescence micrographs of hUSC cells at 4 h of incubation with FITC-labeled nioplexes (Green). (**A**) Uptake of GPxT-CQ at a lipid/DNA ratio of 5:1 (*w*/*w).* 1.25 μg of DNA and 6.25 μg of GPxT-CQ cationic niosome were added to each well; (**B**) uptake with Lipofectamine at a lipid/DNA ratio of 2:1 (*w*/*w*) used as a positive control. 1.25 μg of the plasmid and 2.5 μg lipofectamine were used. Scale bar: 100 μm.

**Figure 5 pharmaceutics-13-00696-f005:**
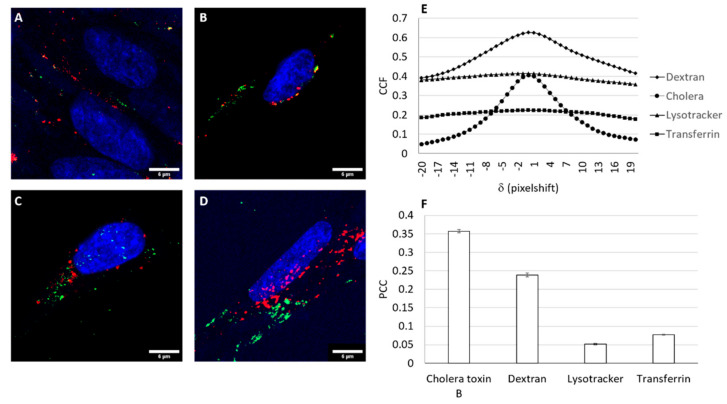
Endocytic and intracellular trafficking pathway detection assay of GPxT-CQ nioplexes in hUSCs. (**A**–**D**) Confocal microscopy merged images showing hUSCs co-incubated with GPxT-CQ nioplexes containing the FITC-labeled pCMS-EGFP plasmid (green) and the endocytic vesicle marker (red): (**A**) caveolae-mediated endocytosis (cholera toxin B), (**B**) macropinocytosis (dextran), (**C**) lysosomes (Lysotracker), and (**D**) clathrin-mediated endocytosis (transferrin). Scale bar: 6 µm. (**E**) Cross-correlation function (CCF) of colocalization between red and green signals. Data are represented as the mean ± SEM, *n* = 3. (**F**) Pearson correlation coefficient (PCC) of red and green signals determined by cross-correlation analysis in each case. Data are represented as the mean ± SEM, *n* = 3. For interpretation of the references to color in this figure legend, the reader is referred to the web version of this article.

**Table 1 pharmaceutics-13-00696-t001:** Percentage of positive cells for each cell surface marker studied by flow cytometry. CD73, CD90, and CD105 correspond to antigens that are present specifically in mesenchymal stem cells. However, CD14, CD34, and CD45 are absent in MSCs. The different fluorophores used were FITC, APC, PE, and AlexaFluor488.

Cell Type	CD73-AlexaFluor488	CD90-APC	CD105-PE	CD14-FITC	CD34-APC	CD45-PE
Commercial mesenchymal stem cells	63.4%	81.7%	70.1%	0.8%	1.3%	20.5%
hUSCs	78.6%	86.4%	66.1%	6.8%	13.4%	52.4%
HEK293	36.5%	0.9%	3.5%	2.8%	0.8%	11.3%

**Table 2 pharmaceutics-13-00696-t002:** Physicochemical characterization of GPxT-CQ cationic niosome and its corresponding nioplex at a cationic lipid:DNA (*w*/*w*) ratio of 5:1. Each value represents the mean ± SD (*n* = 3).

Formulation	Size (nm)	Zeta Potential (mV)	PdI
GPxT-CQ	109.8 ± 1.01	33.4 ± 5.7	0.13 ± 0.01
GPxT-CQ/DNA (5:1)	162.3 ± 2.6	21.2 ± 2.4	0.31 ± 0.04

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
