# Peer review of "Design and Validation of a Process Based on Cationic Niosomes for Gene Delivery into Novel Urine-Derived Mesenchymal Stem Cells"

_pharmaceutics, 2021, doi:10.3390/pharmaceutics13050696_

Round 1

Reviewer 1 Report

de Nanclares and coworkers presented a plasmid delivery platform using niosomes for urine-derived mesenchymal stem cells. The aim of reducing cytotoxicity is important. However, the improvement of biocompatibility severely compromised the delivery efficiency as shown by Figure 3. About 15% transfection efficiency honestly needs improvement. This shall be addressed in the Conclusion. 

  1. Line 94, the authors mentioned that niosomes contain non-ionic surfactants a few lines before, when it incorporate cationic lipids, can it still be defined as niosomes? If the formulation involves cationic lipids, the title may not be appropriate to use "niosomes".
  2. "GPxT-CQ" was never defined in the manuscript. 
  3. The top panel graph in Figure 3 is not properly labeled. Please increase the font and clearly label what the column and the dot represents (please revise this for all the figures in the manuscript). Dosage of transfection reagents and plasmid should be added in the figure caption. 
  4. Line 194~198 should be backed up with proper references. Only looking at colocalization is not convincing enough to indicate the endocytic pathways. The colocalization should be analyzed and quantified with Pearson's correlation coefficient. References regarding pharmacological inhibitors and knockdown approaches should be cited and discussed: DOI: 10.1021/acs.biomac.9b01073; DOI: 10.1021/acsnano.7b02044.  

Reviewer 2 Report

The paper entitled "Design and validation of a gene delivery process based on niosomes into novel urine-derived mesenchymal stem cells" is based urine derived stem cells hUSC, which could have relevant clinical applications like gene therapy.

The study is well designed and of potential clinical use. There are no major comments. Minor English editing can improve the flow of the paper.

Round 2

Reviewer 1 Report

The authors have extensively answered the comments from previous reviewers.